# Functional Link between miR-200a and ELK3 Regulates the Metastatic Nature of Breast Cancer

**DOI:** 10.3390/cancers12051225

**Published:** 2020-05-13

**Authors:** Hyung-Keun Kim, Joo Dong Park, Seung Hee Choi, Dong Jun Shin, Sohyun Hwang, Hae-Yun Jung, Kyung-Soon Park

**Affiliations:** Department of Biomedical Science, College of Life Science, CHA University, Pangyo-Ro 335, Bundang-gu, Seongnam-si 463-400, Gyeonggi-do, Korea; zzbeee@naver.com (H.-K.K.); wnehd06@naver.com (J.D.P.); raru1201@naver.com (S.H.C.); sjun_11@hanmail.net (D.J.S.); blisssulwin@chamc.co.kr (S.H.)

**Keywords:** ELK3, miR-200a, 3’ untranslated region (UTR), post-transcriptional modification, cell migration and invasion, extravasation, breast cancer

## Abstract

Triple-negative breast cancer (TNBC) refers to breast cancer that does not have receptors for estrogen, progesterone, and HER2 protein. TNBC accounts for 10–20% of all cases of breast cancers and is characterized by its metastatic aggressiveness, poor prognosis, and limited treatment options. Here, we show that the metastatic nature of TNBC is critically regulated by a functional link between miR-200a and the transcription factor ELK3. We found that the expression levels of miR-200a and the *ELK3* mRNA were negatively correlated in the luminal and TNBC subtypes of breast cancer cells. In vitro experiments revealed that miR-200a directly targets the 3’ untranslated region (UTR) of the *ELK3* mRNA to destabilize the transcripts. Furthermore, ectopic expression of miR-200a impaired the migration and invasion of TNBC cells by reducing the expression level of the *ELK3* mRNA. In in vivo studies, transfection of MDA-MB 231 cells (a claudin-low TNBC cell type) with exogenous miR-200a reduced their extravasation into the lung during 48 h after tail vein injection, and co-transfection of the cells with an expression plasmid harboring ELK3 that lacked an intact 3’UTR recovered their extravasation ability. Overall, our findings provide evidences that miR-200a and *ELK3* is functionally linked to regulate invasive characteristics of breast cancers.

## 1. Introduction

Post-transcriptional modification of mRNAs is crucial for gene regulatory mechanisms and plays an important role in the control of carcinogenesis by regulating the expression of oncogenes and tumor suppressor genes [1]. The coding sequences of an mRNA is linked to the 5’ and 3’ untranslated regions (UTRs), which are often referred to as cis-acting elements to regulate gene expression at post-transcriptional stage [2]. After being regarded as accessory sequences for a long time, accumulating evidences show that UTRs control mRNA stability, localization, and translation efficiency [3]. Particularly, the sequences of 3’-UTR have been reported to play a pivotal role to control the stability of mRNA by being targeted by trans-acting factors including miRNAs [4]. MicroRNAs (miRNAs) are ~22 nucleotide-long RNA molecules that control gene expression by targeting 3’UTRs to destabilize mRNAs or inhibit their translation [5]. Dysregulation of gene expression by miRNAs can be caused various human diseases, including cancer [6], and miRNAs can act as tumor suppressors or promoters during carcinogenesis. The role of the miR-200 family (miR-200a, miR-200b, miR-200c, miR-141, and miR-429) in epithelial to mesenchymal transition (EMT) and metastasis formation in different tumor types has been studied extensively [7,8,9,10]. The miR-200 family suppresses the metastasis of cancers by targeting aberrantly expressed EMT-related transcription factors. Zinc finger E-box binding homeobox 1 (ZEB1) is the most well-characterized EMT regulator, and the miR-200 family/ZEB1 axis is recognized as a therapeutic target to prevent the progression and invasion of cancers [11,12]. However, the full range of cancer- and metastasis- related targets of miR-200a has not been identified and is likely to differ between different tumor types.

ELK3 is a ternary complex factor that belongs to the ETS family of transcription factors and binds to specific DNA regions at purine-rich GGA core sequences [13]. ELK3 normally functions as a transcriptional repressor but can be converted to a transcriptional activator by the Ras/ERK signaling pathway [14]. Recently, several reports have suggested that ELK3 functions as an oncogene to regulate EMT-mediated metastasis in various cancers, including liver and breast cancers [15,16,17]. In our previous studies, ELK3 expression was significantly correlated with cell migration and invasion in breast cancer [18], and ZEB1 regulated the transcriptional activity of ELK3 to repress E-cadherin expression in breast cancer cells [19]. In addition, ELK3 expression is suppressed by some miRNAs, such as miR-135a and miR-155–5p, to control cancer cell proliferation and angiogenesis [20,21]. However, the mechanisms underlying the regulation of ELK3 expression during breast cancer metastasis are still ambiguous.

We showed previously that the expression levels of the ELK3 mRNA and protein differ between subtypes of breast cancer; specifically, ELK3 expression is higher in the aggressive claudin-low type of triple-negative breast cancer (TNBC) than in the luminal type [18]. Here, as a follow-up study, we examined the molecular mechanisms underlying these differences and found that the expression level of the *ELK3* mRNA in breast cancer subtypes is mainly determined by miR-200a. Our findings indicate that miR-200a targets the 3’UTR of the *ELK3* mRNA directly to ensure post-transcriptional repression of the gene. Ectopic expression of miR-200a in TNBC cells in vitro inhibited cell migration and invasion via destabilization of the *ELK3* mRNA. Furthermore, transient expression of miR-200a in MDA-MB 231 cells inhibited their extravasation into the lung in mice injected via the tail vein, and co-transfection of the MDA-MB 231 cells with miR-200a and the *ELK3* mRNA lacking a miR-200a target site rescued their impaired extravasation ability. Overall, our findings reveal a novel functional link between miR-200a and ELK3 that critically impacts the tumor invasion phenotypes of breast cancers.

## 2. Results

### 2.1. Stability of the ELK3 mRNA Differs between Breast Cancer Subtypes

As reported previously [15], the *ELK3* mRNA was expressed at higher levels in MDA-MB 231 cells (claudin-low TNBC subtype) than in MCF7 cells (luminal breast subtype) (Figure 1A). To understand the mechanisms underlying the difference of *ELK3* mRNA level in breast cancer subtype, we first examined the methylation status of the *ELK3* gene [22]. Compared with those in MDA-MB 231 cells, the CpG islands in the first intron of the *ELK3* gene were more highly methylated in MCF7 cells (Figure 1B). However, the expression level of the *ELK3* mRNA was not changed significantly following demethylation of the gene by the treatment of 5’-azacytidine, which inhibits DNA methyltransferase activity (Figure 1C). Subsequently, we examined the stability of the *ELK3* mRNA in MDA-MB 231 and MCF7 cells using the transcription inhibitor actinomycin D. The stability of the *ELK3* mRNA was two-fold lower in MCF7 cells than in MDA-MB 231 cells at 480 min after actinomycin D treatment (Figure 1D), suggesting that the difference in *ELK3* mRNA expression between MDA-MB 231 and MCF7 cells results from the difference in *ELK3* mRNA stability.

### 2.2. 3’UTR Is Responsible for Destabilization of the ELK3 mRNA

The stability of mRNAs is controlled by elements in their 3’ UTRs [23]. To investigate the role of the 3’UTR in the stability of the *ELK3* mRNA, we generated a plasmid expressing exogenous *ELK3* lacking an intact 3’UTR, and examined the stabilities of the exogenous (exo-) and endogenous (endo-) *ELK3* mRNAs in transiently transfected MCF7 cells using specific primer sets. The expression level of the exo-form of the *ELK3* mRNA was stably maintained up to 8 h after actinomycin D treatment, whereas that of the endo-form was decreased by 50% at 2 h after actinomycin D treatment (Figure 2A). These results suggest that the 3’UTR of the *ELK3* mRNA is associated with its destabilization in MCF7 cells. To confirm this proposal, we generated a luciferase reporter plasmid containing the 3’UTR of *ELK3*, and examined the luciferase activity in transfected MDA-MB 231 and MCF7 cells. As expected, the luciferase activity was significantly higher in MDA-MD 231 cells than in MCF7 cells (Figure 2B).

### 2.3. miR-200a Promotes Destabilization of the ELK3 mRNA by Targeting Its 3’UTR

As well-known regulators of tumor suppressor genes and oncogenes, miRNAs can control the progression of various cancers by binding directly to 3’UTRs to regulate the level and stability of their target mRNAs [5]. Here, we used the TargetScan and starBase databases to identify miRNAs that regulate stability of the *ELK3* mRNA, and selected eight miRNAs (miR-93-5p, miR-106-5p, miR-141-3p, miR-200a-3p, miR-200b-3p, miR-200c-3p, miR-429, and miR-454-3p) with expression levels that were negatively associated with those of the *ELK3* gene (Appendix A). Subsequently, we co-transfected 293T cells with the eight miRNAs and a luciferase reporter plasmid containing the 3’UTR of *ELK3*. Seven of the miRNAs (miR-93-5p, miR-200a-3p, miR-200b-3p, miR-200c-3p, miR-429, and miR-454-3p) reduced the luciferase activity in the co-transfected cells (Appendix A). Next, we transfected MDA-MB 231 cells with the candidate miRNAs and found that only miR-200a suppressed the endogenous *ELK3* mRNA level significantly (Figure 3A), suggesting that miR-200a might be the main factor that determines the amount of *ELK3* mRNA in breast cancer cells. 

Subsequently, we examined the expression levels of miR-200a in breast cancer cell lines, and found that it was highly expressed in MCF7 cells but barely detectable in MDA-MB 231 cells (Figure 3B). Next, we determined whether miR-200a targets the 3’UTR of the *ELK3* mRNA directly to regulate the transcriptional activity of the gene. To this end, we generated a luciferase reporter plasmid containing the 3’UTR of the *ELK3* mRNA with mutated miR-200a target sequences, and examined the luciferase activity in MDA-MB 231 cells co-transfected with a miR-200a mimic and the wild-type or mutated *ELK3* 3’UTR reporter plasmid. As shown in Figure 3C, miR-200a reduced the luciferase activity in cells expressing the wild-type *ELK3* 3’UTR reporter plasmid significantly, but did not affect that in cells expressing the mutated 3’UTR reporter. These findings indicate that miR-200a negatively regulates the stability of the *ELK3* mRNA by targeting its 3’UTR directly.

### 2.4. miR-200a Expression Negatively Correlates with that of the ELK3 mRNA Level in Breast Cancer Cells

To extend our understanding of the negative association between miR-200a and ELK3 in breast cancer, we analyzed the expression patterns of miR-200a, and the *ELK3* mRNA in other breast cancer cells and human breast cancer patient samples. A bioinformatics analysis of ELK3 and miR200a data in a previous report [24] revealed high expression of *ELK3* and low expression of miR-200a in claudin low-type TNBC cells, including MDA-MB 231 and Hs578T cells. On the other hand, luminal-type breast cancer cells, including MCF7 cells, showed opposing patterns of *ELK3* and miR-200a expression (Figure 4A). Next, we used published data from The Cancer Genome Atlas (TCGA) to analyze breast cancer patient samples and found that miR-200a expression was negatively correlated with that of the *ELK3* mRNA in 747 breast cancer patient samples. This negative correlation was found in 415 luminal breast cancer-type samples and 118 TNBC-type samples (Figure 4B). These findings suggest that miR-200a is a negative regulator of the *ELK3* mRNA level in breast cancer.

### 2.5. miR-200a/ELK3 Axis Regulates Cell Invasion and Extravasation in Breast Cancer

Our previous studies demonstrated that *ELK3* plays an essential role in inducing the migration and metastasis of breast cancer cells such as MDA-MB 231 cells [15,18]. Therefore, we investigated whether the miR-200a/*ELK3* axis controls the metastasis of breast cancer cells in vitro. MDA-MB 231 cells were transfected with a miR-200a mimic to suppress the *ELK3* mRNA level. Ectopic expression of miR-200a inhibited the migration and invasion of MDA-MB 231, and co-transfection of miR-200a and expression plasmid encoding *ELK3* that lacked an intact endogenous 3’UTR (pcDNA3.1- Flag-*ELK3*) restored the phenotypes (Figure 5A). Consistent with the cell migration and invasion phenotype, the *ELK3* mRNA level and protein level was suppressed by transfection of MDA-MB 231 cells with a miR-200a mimic, and was recovered to that of the control group by co-transfection with pcDNA3.1-Flag-*ELK3* (Figure 5B,C). Similar to MDA-MB 231, the migration and invasion of Hs578T, another triple negative breast cancer cell line to express high level of ELK3, was affected by the expression of miR-200a and miR-200a/pcDNA3.1-Flag-*ELK3* (Appendix A). To further examine the molecular mechanisms involved in the regulation of cell migration and invasion by the miR-200a/*ELK3* axis, we analyzed the gene expression profiles of miR-200a alone or miR-200a plus pcDNA3.1-Flag-*ELK3* compared with negative control in MDA-MB 231 cells. An analysis of differentially expressed genes (DEGs) showed that the negative control group and miR-200a plus *ELK3* group had similar genetic signatures compared with the miR-200a alone group (Figure 5D). Gene Ontology (GO) analysis reveals that 99 DEGs were significantly enriched in membrane and extracellular-related components, and cell migration-related pathways (Figure 5E). Notably, STRING analysis reveals that CDH1 is a hub of 99 DEGs (Figure 5F). Combining with a previous study that ELK3 directly represses *CDH1* expression in MDA-MB 231 [19], this result indicates that CDH1 might be the main target of miR-200a/*ELK3* axis to regulate migration and invasion of cancer cells. To validate the speculation, we performed qRT-PCR analysis of 8 genes including *CDH1* (data not shown). Along with *CDH1*, *Lama5* (*Lamininα5*) and *TNS1* (*Tensin1*), which encode extracellular proteins, was confirmed to be regulated by miR-200a/*ELK3* axis (Figure 5G). 

Next, we determined whether the miR-200a/*ELK3* axis regulates breast cancer metastasis in vivo. To this end, we performed an extravasation analysis of mice that were injected via the tail vein with control MDA-MB 231 GFP-luciferase cells, or those expressing miR-200a alone or miR-200a plus pcDNA3.1-Flag-*ELK3*. An examination of tumor cell extravasation 48 h after the tail vein injection revealed that cancer cells were detected in the lungs of the control group and miR-200a plus *ELK3* group (Figure 6A,B). By contrast, the number of extravasated cells in the lung was significantly lower in the group injected with MDA-MB 231 GFP-luciferase cells expressing miR-200a alone (Figure 6A,B). Overall, our findings suggest that the miR-200a/*ELK3* axis regulates the metastatic characteristics of TNBC cells both in vitro and in vivo.

## 3. Discussion

ELK3 expression is associated with malignancy of various cancers including TNBC, and is expected to be a putative molecular target to prevent cancer progression [16,19,25,26]. However, the molecular mechanisms involved in the regulation of *ELK3* mRNA expression during breast cancer metastasis are still unclear. Here, we report that miR-200a is a major determinant of the *ELK3* mRNA level in different breast cancer subtypes, and that the miR-200a/*ELK3* axis is functionally linked to the metastatic characteristics of TNBC cells. An in silico analysis of human breast cancer patient samples and breast cancer cells revealed a negative relationship between miR-200a and *ELK3* expression. Consistently, in vitro and in vivo analyses demonstrated that miR-200a targets the 3’UTR of the *ELK3* mRNA directly. Furthermore, we found that ectopic expression of miR-200a blocked the *ELK3*-mediated invasion and extravasation of TNBC cells. 

DNA methylation is a mechanism involved in the regulation of gene expression [27]. Cancer cells are characterized by dysregulated DNA methylation profiles with mutations or abnormal expression of epigenetic regulators, such as Ten-eleven translocation (TET) proteins, and major epigenetic changes caused by those proteins play a critical role to control transcriptional activity of oncogenes as well as tumor suppressor genes [28]. 

Although we found that CpG islands located in the first intron of the *ELK3* gene were more highly methylated in MCF7 cells than in MDA-MB 231 cells, the *ELK3* mRNA level in MCF7 cells was not increased by treatment with 5’-azacytidine at a concentration known to induce the transcription of other methylated genes, such as *BRF2* [29]. One possible explanation for this finding is that the transcriptional activator of *ELK3* is limited in MCF7 cells. Recently, ZEB1 has been reported as a transcriptional activator of *ELK3* [19]. Therefore, the lack of effect of 5’-azacytidine on *ELK3* mRNA levels might be attributed to low expression of ZEB1 in MCF7 cells. 

Our results suggest that, in the luminal and TNBC subtypes of breast cancer, the *ELK3* mRNA level is mainly regulated at the post-transcriptional stage by miRNAs, rather than by DNA methylation. Specifically, our data revealed that miR-200a is a major post-transcriptional regulator of *ELK3* that determines its expression pattern in breast cancer cells. 

Based on the role of miR-200a to target EMT-related transcription factors, miR-200a has been recognized as a therapeutic target to prevent cancer progression [7,8,9,10,11,12]. Recently, it is reported that ZEB1, a representative target of miR-200a, collaborates with ELK3 to repress the *Cdh1* expression in TNBC cells [19]. Considering that ELK3 functions as a master regulator to orchestrate invasion and metastasis of breast cancers [15,18], our data that miR-200a regulates *ELK3* expression by targeting to 3’-UTR of *ELK3* mRNA is expected to expand our understanding about cancer progression mechanism of different breast cancer subtypes.

It is well-known that *ZEB1* and miR-200a forms feedback loop, and the miR-200a/*ZEB1* axis has been recognized as a determinant of cellular plasticity during cancer progression [30,31,32,33]. Our data also showed that *ZEB1* mRNA level was decreased after miR-200a transfection, however, introduction of *ELK3* did not rescue the decreased expression of *ZEB1* mRNA level in miR-200a transfected cells (Appendix A), suggesting that ELK3 does not function as a transcriptional activator of *ZEB1*. Interestingly, we found that miR-200a/*ELK3* axis regulates the expression of *Lama5* (*Lamininα5*) and *TNS1* (*Tensin1*), which encode extracellular proteins [34]. This founding provides the possibility that miR-200a negatively regulates ELK3-mediated cell migration and invasion through extracellular matrix remodeling. Our finding is consistent with previous reports that one of major function of microRNAs (miRNAs) is to regulate extracellular matrix-related gene to control cancer invasiveness [35,36]. Notably, in our current study, the phenotypes impaired by overexpression of miR-200a were rescued by co-expression of exogenous *ELK3* transcripts. These results support the concept that *ELK3* is able to regulate the metastatic characteristics of TNBC cells independent of *ZEB1*, at least in the MDA-MB 231 and Hs578T cell lines. 

The biological role of miR-200a in metastasis is still controversial and appears to depend on its specific target genes. One report showed that miR-200a promotes breast cancer metastasis by targeting *YAP1* [37]. In another study, extracellular vesicles, including miR-200, increased breast cancer metastasis [38]. On the other hand, several groups have found that miR-200a suppresses the expression of genes related to EMT, resulting in the inhibition of cancer metastasis [39,40]. For example, miR-200a inhibits cell migration and invasion by targeting different genes, such as GAB1 and EPHA2 [41,42]. Based on our in vitro and in vivo data, we propose that *ELK3* is a novel target of miR-200a. We suggest that miR-200a functions as a suppressor of metastatic characteristics by targeting *ELK3* in breast cancers, and that the miR-200a/*ELK3* axis determines the fate of breast cancer cells whether they are undergoing cell migration, invasion, or metastasis. To our knowledge, this report is the first to demonstrate that miR-200a and *ELK3* are functionally linked to regulate the metastatic nature of breast cancer.

## 4. Materials and Methods 

### 4.1. Cell Culture and Transfection

The human breast cancer cell lines MCF7 and MDA-MB 231 and the human kidney cell line HEK293T were obtained from the American Type Culture Collection (ATCC, Manassas, VA, USA) and were cultured in Dulbecco’s modified Eagle’s medium (Gibco, Grand Island, NY, USA) supplemented with 10% fetal bovine serum (Gibco) and 1% penicillin-streptomycin (Gibco). The human breast cancer cell line Hs578T was cultured in Dulbecco’s modified Eagle’s medium (Gibco) supplemented with 10% fetal bovine serum (Gibco), 0.01 mg/ml insulin (Sigma, St. Louis, MO, USA), and 1% penicillin-streptomycin (Gibco). All cell lines were incubated at 37 °C and 5% CO_2_. Target cells were transfected with miR-200a (100 nM) and pcDNA3.1-Flag-*ELK3* (30 ng) using Lipofectamine 3000 (Invitrogen, Carlsbad, CA, USA). 

### 4.2. Plasmids and miR-200a 

All plasmids, including pcDNA3.1-Flag-*ELK3* [17], were generated from cDNA by PCR. The protein coding region of *ELK3* was ligated into the pcDNA3.1 vector (Invitrogen). The 5’UTR and 3’UTR of the *ELK3* mRNA were obtained from the NCBI platform (NM_005230.4). For the luciferase assay, pGL3/*ELK3*–3’UTR and pGL3/*ELK3*–3’UTR mutant DNA constructs were generated by PCR. The mutant plasmid was constructed using a site-directed mutagenesis kit (Agilent, Santa Clara, CA, USA). All primers are shown in Table 1. The miR-200a mimic and the non-specific control miRNA mimic were synthesized by Genolution Inc. (Songpa, Seoul, South Korea). The sense and antisense target sequences of miR-200a are 5’-UAA CAC UGU CUG GUA ACG AUG U-3’ and 5’-ACA UCG UUA CCA GAC AGU GUU A-3’, respectively. 

### 4.3. Quantitative RT-PCR

Total RNA was extracted from breast cancer cell lines using TRIzol (Invitrogen), according to the manufacturer’s protocol, and cDNA was synthesized from 1 μg of each total RNA using the LeGene Premium Express First-Strand cDNA Synthesis System (Legene Biosciences, San Diego, CA, USA). To quantify mRNA expression, *GAPDH* was used for normalization of RT-PCR and real-time quantitative PCR data. Real-time quantitative PCR was performed using TOPreal^TM^ qPCR 2X PreMIX (Enzynomics, Daejeon, Chungnam, South Korea) and the CFX Connect Real-Time PCR Detection System (Bio-Rad, Hercules, CA, USA). The HB miR Multi Assay Kit was used to detect miRNAs (HeimBiotek, Pangyo, Gyeonggi, South Korea). The levels of miR-200a in MDA-MB 231 and MCF-7 samples were normalized to those of RNU6B. 

### 4.4. Luciferase Reporter Assays

For luciferase assays, cells were transfected with the pRL-TK vector and pGL3-promoter vectors containing the 3’UTR of *ELK3* (Promega, Madison, WI, USA). At 48 h post-transfection, luciferase activity was measured using the Dual-Luciferase Reporter Assay Kit (Promega), according to the manufacturer’s protocol. The level of firefly luciferase was normalized to that of Renilla luciferase. 

### 4.5. Bisulfite Sequencing

Genomic DNA was extracted using a genomic DNA extraction kit (Bioneer, Daejeon, Chungnam, South Korea) and modified with sodium bisulfite using the Imprint DNA Modification Kit (Sigma). The modified DNA was amplified by primers spanning the +814 to +1041 region of *ELK3*. The PCR product was purified with a PCR purification kit (Bioneer) and cloned into a T&A cloning vector (RBC Bioscience, New Taipei City, Taiwan). Each clone was selected by ampicillin and extracted using the Nano-Plus Plasmid Mini Extraction kit (Bioneer). Five positive clones were sequenced, and the numbers of total and methylated CpG dinucleotides were determined.

### 4.6. Prediction of Candidate miRNAs

To predict miRNAs targeting the *ELK3* mRNA, we first performed computational screening using TargetScan v7.2 (http://www.targetscan.org/vert_72/). Correlations between the expression levels of candidate miRNAs and those of *ELK3* were analyzed in invasive breast cancer patient data from starBase v3.0 (http://starbase.sysu.edu.cn/index.php). In addition, we also used breast cancer expression data from GEO DataSet GSE41313. Statistical analysis was performed using GraphPad Prism 5 (GraphPad Prism, La Jolla, CA, USA). 

### 4.7. Cell Migration and Invasion Assay 

For the migration assay, cells were plated onto the inserts of Transwell plates (Corning Inc., Corning, NY, USA) and maintained in serum-free media. The bottom chamber contained medium supplemented with 10% fetal bovine serum as a chemoattractant. After incubation for 18 h, the cells were fixed with 4% paraformaldehyde (Elbio, Pangyo, Gyeonggi, South Korea), and the Transwell inserts were stained with crystal violet (Sigma). For the invasion assay, the Transwell inserts were coated with Matrigel (Corning Inc.) and then treated as described for the migration assay. After 24 h, invasive cells were analyzed by crystal violet staining. 

### 4.8. Mouse Lung Extravasation Assays

Female athymic nude mice were purchased from Koatech (Pyeongtaek, Gyeonggi, South Korea). MDA-MB 231 GFP-luciferase cells (1 × 10^6^ cells per mouse) were injected into the tail vein of mice (*n* = 7 per group). After 48 h, the lungs were harvested and the GFP signal was analyzed using a fluorescence-labeled organism bioimaging instrument (FOBI) system (Neo-Science, Suwon, Gyeonggi, South Korea). Lung tissues were embedded with OCT compound (Corning Inc., Corning NY, USA) for frozen sectioning. To analyze the extravasation of tumor cells, GFP-positive cells were counted under a fluorescence microscope (ZEISS, Oberkochen, Germany). All animal care and experiments were performed under approved animal protocol by Institutional Committee for Use and Care of Laboratory Animals of laboratory animal research center in CHA University (IACUC-200011). 

### 4.9. RNA-Seq Analysis

For gene expression profiling, libraries were prepared using the QuantSeq 3’mRNA-Seq kit (Ebiogen, Seoul, South Korea). Sequencing libraries were generated replicate in regions close to the 3’ end of polyadenylated RNAs. Gene expression patterns were analyzed using ExDEGA software (Excel Based Differentially Expressed Gene Analysis, Ebiogen). Using this approach, we selected 1539 genes with *p* < 0.05, fold-change > 1.5, and normalized data > 3. The RNA-seq data reported in this article has been deposited in NCBI’s Gene Expression Omnibus (GEO) and are accessible through GEO Series accession number: GSE148562. Differentially expressed genes were used to generate heat maps using the MeV 4.9 software package with the hierarchical clustering method. For biological function analysis, gene ontology (GO) analysis including cellular component (CC) and biological function (BP) were performed using the Database for Annotation, Visualization, and Integrated Discovery Version 6.9 (DAVID; https://david.ncifcrf.gov/; accessed on March 28, 2020) [43]. To analysis the network between proteins, STRING ver.11 was used [44]. 

### 4.10. Genomic Analyses of Human Breast Cancer Patient Samples and Breast Cancer Cells

To analyze the expression patterns of miR-200a and the *ELK3* mRNA in human breast cancer patient samples, we downloaded RNA-seq data from TCGA using the UCSC Xena tool (http://xena.ucsc.edu/). All miRNA expression data used were log2 (total_RPM+1)-transformed and all RNA-seq data used were log2(x+1)-transformed RSEM-normalized counts. The breast cancer subtype was categorized based on the receptor status using the phenotype data from TCGA. Overall, after removing patients without miRNA expression information, data were obtained for 747 patients. Correlations between the expression levels of miR-200a and the *ELK3* gene in patients were analyzed using Spearman’s linear correlation. In breast cancer cells, the expression levels of *ELK3* and miR-200a were classified as low or high according to the median expression, as described in a previous report [24].

### 4.11. Western Blot Analysis

Cells were washed with phosphate-buffered saline (PBS, Gibco BRL, Waltham, MA, USA) and lysed in cell lysis buffer (Cell Signaling Technology, Danvers, MA, USA). Total cell extracts were resolved by sodium dodecyl sulfate-polyacrylamide gel electrophoresis (SDS-PAGE), transferred to polyvinylidene fluoride membranes (PVDF, Bio-Rad, Hercules, CA, USA), and blotted with antibodies against ELK3 (NBP2-01264, Novus, Littleton, CO, USA) and GAPDH (sc-166574, Santa Cruz Biotechnology, CA, USA). Immunoreactivity was detected with enhanced chemiluminescence (ECL, Thermo Fisher Scientific, Rochester, NY, USA).

### 4.12. Statistical Analysis

Statistical analysis was performed using GraphPad Prism software. *p*-values were determined using Student’s *t*-tests. Error bars represent the standard error of the mean (S.E.M.). Statistical significance was considered as *p* < 0.05. All data used in statistical analyses originated from at least three independent experiments.

## 5. Conclusions

We found that miR-200a determines the expression level of ELK3 in breast cancer subtypes by directly targeting the 3’UTR of *ELK3* mRNA. In vitro and in vivo experiments suggest that miR-200a and ELK3 expression is functionally linked to the metastatic characteristics of aggressive triple-negative breast cancer cells. To the best of our knowledge, the results demonstrate that miR-200a is implicated in the regulation of the oncogenic activity of ELK3 in breast cancers.

## Figures and Tables

**Figure 1 cancers-12-01225-f001:**
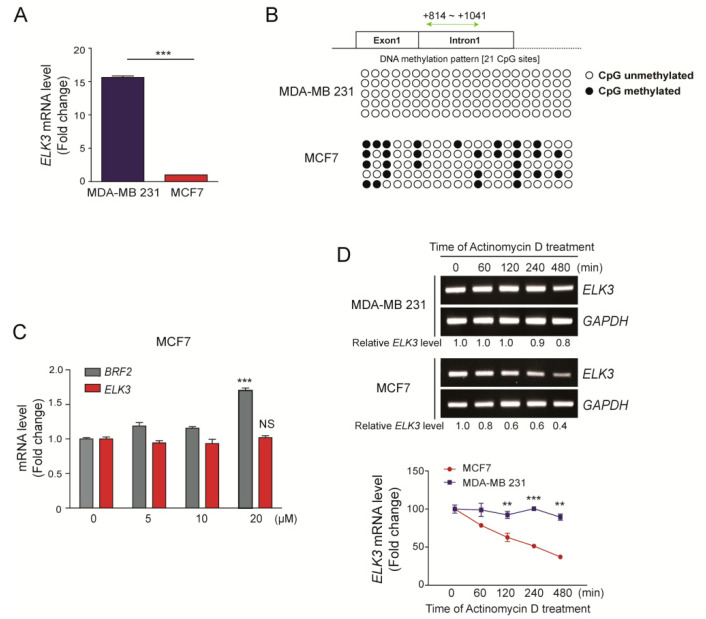
Differences between the expression levels of the *ELK3* mRNA in MDA-MB 231 and MCF7 cells are independent of genomic DNA methylation. (**A**) A quantitative analysis of the *ELK3* mRNA level in MDA-MB 231 and MCF7 cells. Error bars represent the S.E.M.; *** *p* < 0.001 (Student’s *t*-test). (**B**) Methylation pattern of the first intron region of the *ELK3* gene in MDA-MB 231 and MCF7 cells. Methylation of 21 CpG islands in the first intron of *ELK3* (+814 ~ +1041) was analyzed by bisulfite sequencing. White circles indicate unmethylated CpG sites, and black circles indicate methylated CpG sites. (**C**) Effect of 5’-azacytidine treatment of MCF7 cells on expression of the *ELK3* mRNA. MCF7 cells were treated with 5’-azacytidine for 72 h, and *ELK3* expression was analyzed by qRT-PCR. Expression of the *BRF2* gene was analyzed as a positive control for the effect of 5’-azacytidine. Error bars represent the S.E.M.; *** *p* < 0.001 (Student’s *t*-test). (**D**) MDA-MB 231 and MCF7 cells were treated with actinomycin D (2.5 μg/ml) for the indicated times, and the *ELK3* mRNA level was analyzed by RT-PCR and qRT-PCR. The graph represents the *ELK3* mRNA level relative to GAPDH. Error bars represent the S.E.M.; ** *p* < 0.01 and *** *p* < 0.001 (Student’s *t*-test).

**Figure 2 cancers-12-01225-f002:**
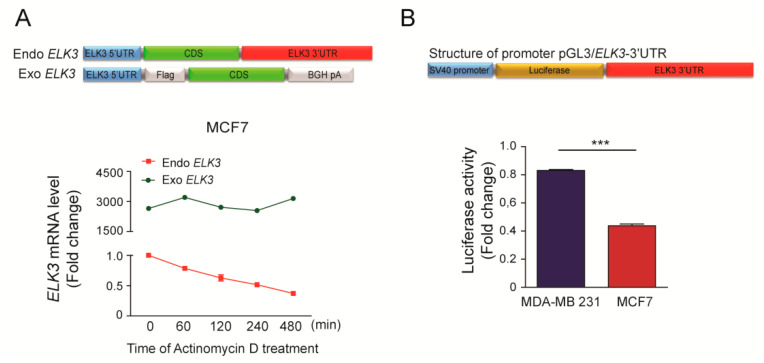
3’UTR determines the stability of the *ELK3* mRNA. (**A**) Stabilities of the endogenous *ELK3* mRNA and an exogenous form lacking the 3’UTR in MCF7 cells. The cells were transfected with the exo-form of ELK3 and treated with actinomycin D (2.5 μg/ml) for the indicated times. The levels of the exo- and endo-forms of the *ELK3* mRNA were analyzed by qRT-PCR. The endo-form was detected with primers targeting the 3’UTR, and the exo-form was detected with primers targeting the region spanning the Flag tag to the coding sequence of *ELK3*. (**B**) The luciferase activity in MDA-MB 231 and MCF7 cells transfected with a luciferase reporter plasmid harboring the 3’UTR of *ELK3*. Error bars represent the S.E.M.; *** *p* < 0.001 (Student’s *t*-test).

**Figure 3 cancers-12-01225-f003:**
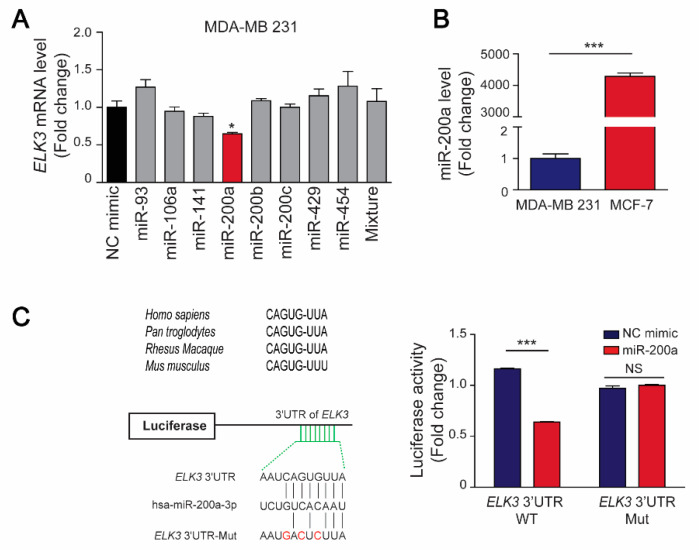
MiR-200a targets the 3’UTR of the *ELK3* mRNA in breast cancer cells. (**A**) The indicated miRNAs (100 nM) were transfected into MDA-MB 231 cells for 48 h, and the *ELK3* mRNA level was analyzed by qRT-PCR. Error bars represent the S.E.M.; * *p* < 0.05 (Student’s *t*-test). (**B**) Quantitative analyses of miR-200a expression in MCF7 and MDA-MB 231 cells. Error bars represent the S.E.M.; *** *p* < 0.001 (Student’s *t*-test). (**C**) Regulation of the luciferase activity of a reporter plasmid containing the wild-type or mutated 3’UTR of the *ELK3* mRNA by miR-200a. *ELK3* 3’UTR Mut indicates mutation of the miR-200a target sequences. Luciferase activity was measured after co-transfection of MDA-MB 231 cells with a miR-200a mimic and the indicated luciferase reporter plasmids for 48 h. Error bars represent the S.E.M.; *** *p* < 0.001 (Student’s *t*-test).

**Figure 4 cancers-12-01225-f004:**
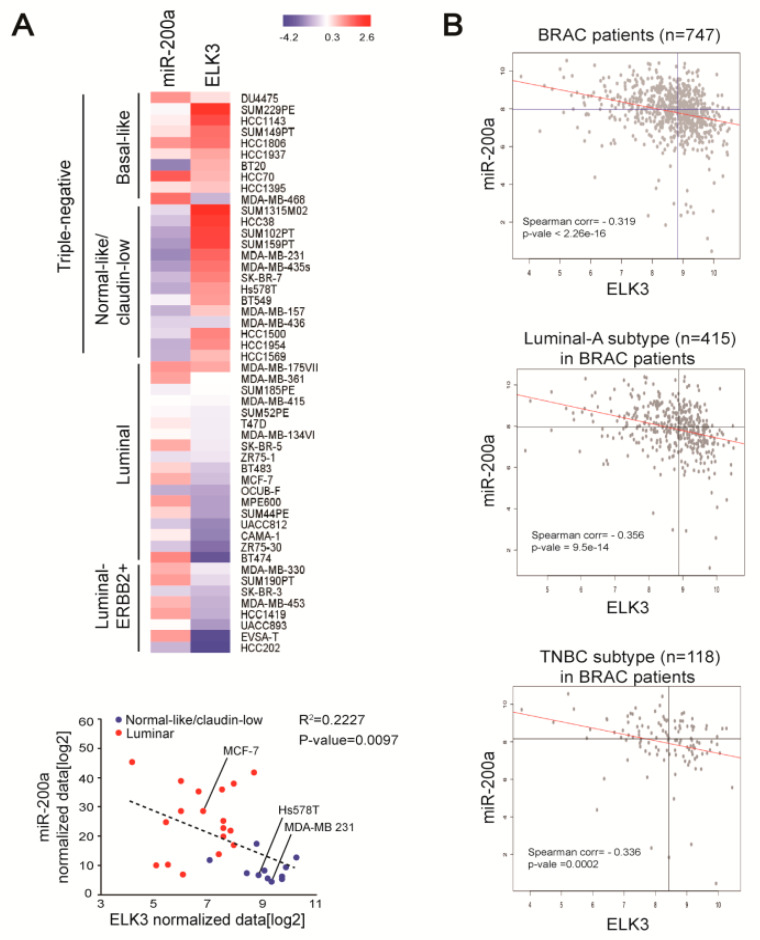
Expression levels of miR-200a and *ELK3* mRNA are negatively correlated in breast cancer cells and patient samples. (**A**) Heat map of miR-200a and *ELK3* mRNA expression in breast cancer cell lines. The blue dots indicate the normal-like/claudin-low breast cancer subtype and the red dots indicate the luminal breast cancer subtype. (**B**) Correlation between miR-200a and *ELK3* mRNA expression in breast cancer patients (*n* = 747). Luminal-A subtype (*n* = 415) and TNBC subtype (*n* = 118) samples were selected based on the receptor expression phenotype. All data were downloaded from the TCGA database.

**Figure 5 cancers-12-01225-f005:**
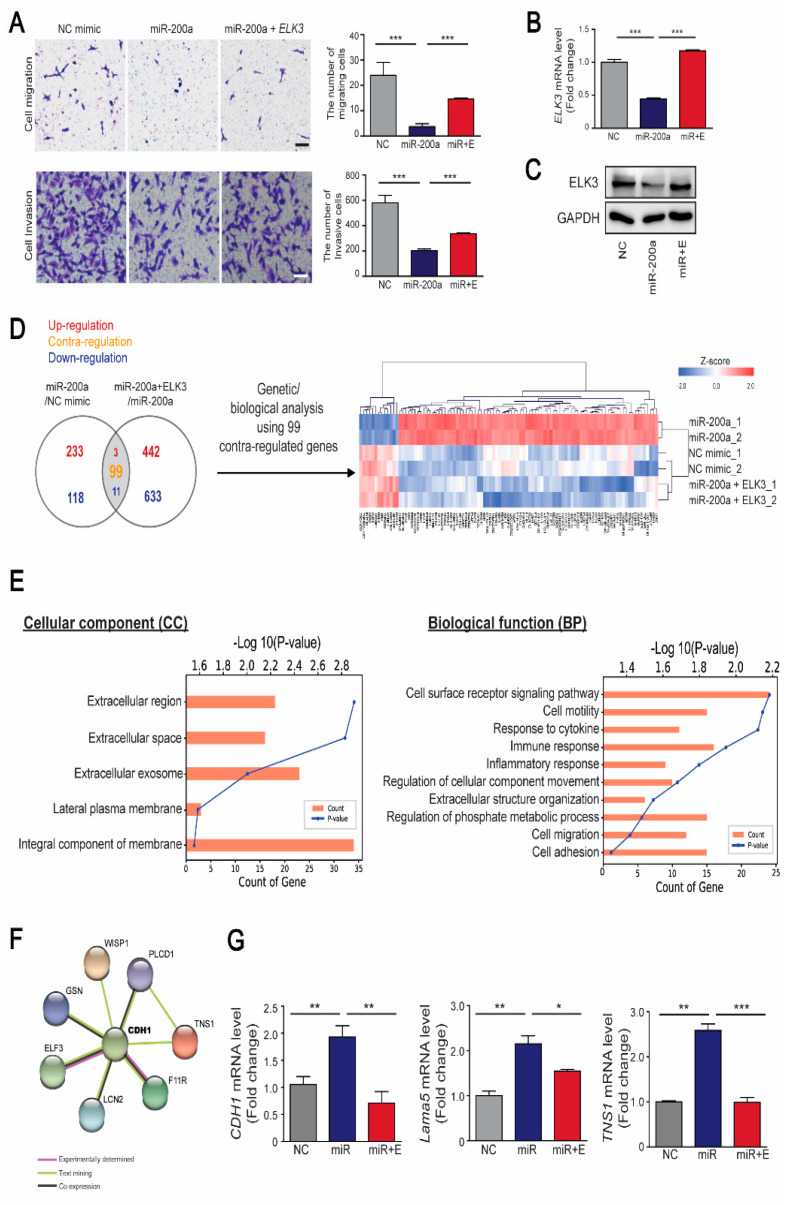
The miR-200a/*ELK3* axis regulates the invasion and migration of TNBC cells in vitro. (**A**) Representative images showing the cell migration and invasion of MDA-MB 231 cells expressing a negative control (NC) mimic, miR-200a (100 nM), or miR-200a (100 nM) plus pcDNA3.1-Flag-*ELK3* (30 ng). The graph indicates quantified cell migration and invasion. Migrating and invasive cells were counted in four randomly selected regions. Scale bars, 20μm. Error bars represent the S.E.M.; *** *p* < 0.001. (**B**) Quantitative analyses of *ELK3* mRNA levels following the indicated treatments. Error bars represent the S.E.M.; *** *p* < 0.001 (Student’s *t*-test). (**C**) The protein level of ELK3 in MDA-MB 231 cells expressing negative control (NC) mimic, miR-200a, or miR-200a plus pcDNA3.1-Flag-*ELK3*. (**D**) Venn diagram of differentially expressed genes (DEGs) in the following groups of MDA-MB 231 cells: those expressing miR-200a, those expressing the NC mimic, and those expressing miR-200a plus pcDNA3.1-Flag-*ELK3*. DEGs were selected based on a cut-off of *p*< 0.05, fold-change > 1.5, and normalized data > 3. A total of 113 DEGs were identified. Among them, 99 DEGs showed contra-regulated expression between the two groups that were compared. (**E**) Gene Ontology (GO) analysis of 99 DEGs represented by significantly enriched cellular component (left panel) and biological function (right panel). (**F**) STRING analysis showing protein–protein interaction of CDH1 and 7 proteins. (**G**) Quantitative analyses of *CDH1*, *Lama5*, and *TNS1* mRNA levels following the indicated treatments. Error bars represent the S.E.M.; * *p* < 0.05, ** *p* < 0.01, and *** *p* < 0.001 (Student’s *t*-test).

**Figure 6 cancers-12-01225-f006:**
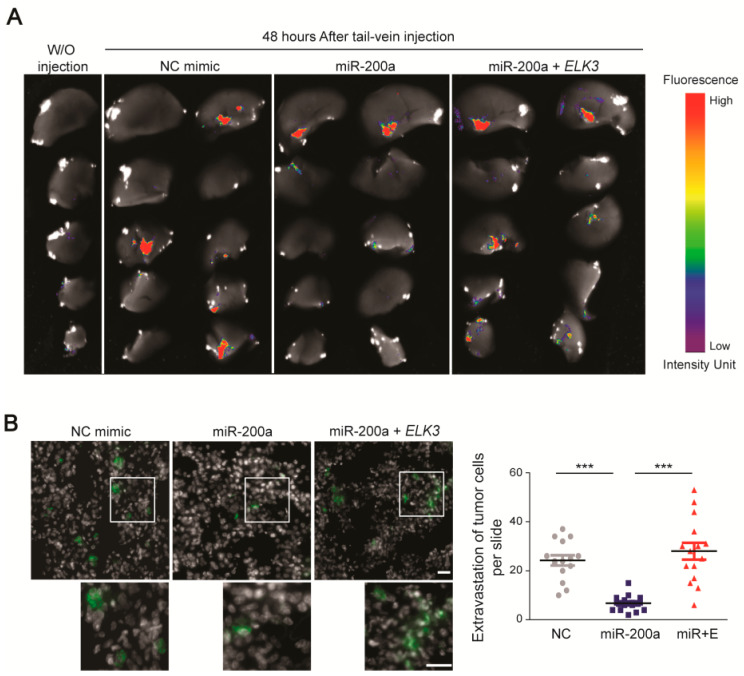
Functional link between miR-200a and *ELK3* regulates the lung extravasation of MDA-MB 231 cells in vivo. (**A**) Fluorescence optical images of the lung 48 h after tail vein injection of mice with control MDA-MB 231 GFP-luciferase cells or those expressing miR-200a alone or miR-200a plus pcDNA3.1-Flag-*ELK3*. “W/O injection” indicates negative control mice that did not receive a tail vein injection. (**B**) Immunofluorescence images of GFP, indicating extravasated tumor cells, in the lungs of mice injected via the tail vein as described for (a). The gray area indicates the nucleus. Scale bars, 20 μm. Extravasation of tumor cells was determined by counting the number of GFP-positive cells. Error bars represent the S.E.M.; *** *p* < 0.001 (Student’s *t*-test).

**Table 1 cancers-12-01225-t001:** Primer lists for qRT-PCR.

Gene	Forward Primer (5’ to 3’)	Reverse Primer (5’ to 3’)
*ELK3*	ACCCAAAGGCTTGGAAATCT	TGTATGCTGGAGAGCAGTGG
*ELK3*_bisulfite	GAGTGGGTAAAGTGTATTGGTGTT	AATCATTTCTTACCTAATCCTTCTCC
*ELK3*_5’UTR	TAGCTAGCAAAAGCCTGTTTACACAGACTGC	TAGCTAGCACCCAGATGTGGGGGAGT
*ELK3*_3’UTR	CTCGAGTGACGTCTGGCCACAATTAAG	GTCGACTTGGTTGAGATTTTTGCACAT
*ELK3*_3’UTR_Mut	TAAGAGTCATTAAGCAGACATAAAAGGGA	GACTCTTAAACTGCTATGGGAAAAGTTTTATAG
*BRF2*	CAGAAGTGGAGACCCGAGAG	CAGGGAGGGTTAGGGACACT
Endo_*ELK3*_CDS_3’UTR	TGATGACGTCTGGCCACAAT	GGTAAACTAGCCCGTGGGG
Exo_Flag_*ELK3*_CDS	TGATGTTCTTGTCATAATAGTATCGCAG	GATTACAAGGATGACGACGATAAGAA
*LAMA5_*	GGACTACATGGGTGTGTCTC	TTTCCTGGATCATCTGTCTC
*TNS1*	GGCTTAGAGCGAGAGAAGCA	CCCGTCCAGAGAAGAGAGTG
*CDH1*	ATGCAGAAACTGGCATCCTC	AGTCCTCGGACACTTCCACT
*ZEB1*	TGCACTGAGTGTGGAAAAGC	TGGTGATGCTGAAAGAGACG
*GAPDH*	GGGTGTGAACCATGAGAA	GTCTTCTGGGTGGCAGTGAT

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
