# Peer review of "Functional Link between miR-200a and ELK3 Regulates the Metastatic Nature of Breast Cancer"

_cancers, 2020, doi:10.3390/cancers12051225_

Round 1
Reviewer 1 Report
All concerns addressed
Author Response
We greatly appreciate the reviewer’s willingness to consider our manuscript publication in Cancers.
Reviewer 2 Report
I would like to thank the authors for their efforts in improving their manuscript. Not all the concerns I raised have been addressed in the revised manuscript, but the authors tried to justify the reasons, which were mainly ascribed to technical problems.
Unfotunately, the quality of Figure 6 remains quite poor. Chip analysis and western blot that were suggested to validate data from RNA-seq have not been performed. RNA-seq data have been now better described and data deposited on GEO. The graphical abstract has been modified with additional details.
Minor concern:
Please, add "data not shown" or add the results when describing that 8 genes, including CDH1, were analyzed by qRT-PCR, as only 2 targets are represented in Figure 5F (line 201).
Author Response
I would like to thank the authors for their efforts in improving their manuscript. Not all the concerns I raised have been addressed in the revised manuscript, but the authors tried to justify the reasons, which were mainly ascribed to technical problems.
=>We greatly appreciate the reviewer’s careful evaluation of our revised manuscript and kind willingness to accept our justification.
Unfortunately, the quality of Figure 6 remains quite poor.
=>We agree to the reviewer’s concern. Due to the limitation of our fluorescence analysis equipments of our institution, however, we are not able to improve the image quality any further. Would you kindly understand our limitation?
Chip analysis and western blot that were suggested to validate data from RNA-seq have not been performed.
=>We apologize for our inattention to miss the reviewer’s comments.
(1) Previously, the reviewer suggested us to validate the DEGs by RT-qPCR and western blot.
Following the reviewer’s suggestion, we validated CDH1, Lama5 and TNS1 as target genes regulated by miR200a/ELK3 axis by presenting RT-qPCR data (Figure 5G). We also tried immunoblot, but we had difficulty detecting these proteins by immunoblot, even from control samples. We consider that these problems are caused by low expression of these 3 genes in mesenchymal cells like MDA-MB231.
(2) The reviewer also suggested us to perform ChIP analysis of ELK3 binding to at least to Cdh1 target gene before and after miR200a transfection.
Indeed, we published the data to demonstrate the direct binding of ELK3 to Cdh1 promoter (Cho HJ et al. Molecular Cancer Research. 2019). Since the qualified ELK3 antibody suitable for ChIP analysis is not commercially available yet, we demonstrated ELK3 binding to Chd1 promoter by point mutation of ELK3 binding motif on Cdh1 promoter as below.
- Cho HJ et al. ZEB1 collaborates with ELK3 to repress E-cadherin expression in triple-negative breast cancer cells. Molecular Cancer Research. 317(6):C1128-C1142.
RNA-seq data have been now better described and data deposited on GEO. The graphical abstract has been modified with additional details.
=> We greatly appreciate the reviewer’s comments. We believe that our manuscript has been improved with detailed analysis of RNA-seq data and modified graphical abstract.
Minor concern:
Please, add "data not shown" or add the results when describing that 8 genes, including CDH1, were analyzed by qRT-PCR, as only 2 targets are represented in Figure 5F (line 201).
=>Thank you for your comment. Following the reviewer’s suggestion, we added ‘data not shown’ at line 201.

Reviewer 3 Report
This manuscript is not for publication
It needs major revision
Abstract is not clear nor concise
Role of miR-200a and ELK3 in the cancer should be clearly explained
In Abstract concussion should be stated exactly what Author finding is “miR-200a promotes degradation of the ELK3 mRNA in the 3’UTR
In the Introduction and Discussion shod be more elaborated about untranslated region (UTRs) and role and action of miR-200a and ELK3 on that region related for cancer progression. Methylation process should be explained in details, Triple negativity of breast cancer should be elaborated in details
In section of the Results if Authors use 5’-azacytidine for example they should explain in details role of the compound and present thet in the text or in the Fig 1 legend
Figure 3 and Figure 5 should be recombined in one figure and show together
miR-200a and ELK3, rest of unnecessary figures should be excluded
Figure 5. Unnecessary figures should be excluded especially FGH parts because not clear , what Authors exactly want to show
Figure 6. Unnecessary figures should be excluded
Author Response
- This manuscript is not for publication. It needs major revision.
=> As we defended previously, our work provides the novel insight to understand the mechanisms underlying the miR200a/ELK3 axis that regulates extravasation during breast cancer metastasis. Such as above reasons, we believe that this study will be of considerable interest to the readers of Cancers.
- Abstract is not clear nor concise. Role of miR-200a and ELK3 in the cancer should be clearly explained.
=> To describe the role of miR-200a/ELK3 axis in the cancer more clearly, we revised conclusion paragraph in the abstract
(line 28-30) Overall, our findings provide evidences that miR-200a and ELK3 is functionally linked to regulate invasive characteristics of breast cancers
In Abstract concussion should be stated exactly what Author finding is “miR-200a promotes degradation of the ELK3 mRNA in the 3’UTR
=> Following the reviewer’s comment, we revised the paragraph in the abstract as below;
(line 22-23) In vitro experiments revealed that miR-200a directly targets the 3’-untranslated region (UTR) of the ELK3 mRNA to destabilize the transcripts.
- In the Introduction and Discussion shod be more elaborated about untranslated region (UTRs) and role and action of miR-200a and ELK3 on that region related for cancer progression.
=>The importance of 5’ to 3’ UTR in the post-transcriptional regulation of gene expression is described at line 39-40 of Introduction.
The biological meaning of miR-200a/ELK3 axis for breast cancer progression is discussed at line 305-316 in the manuscript.
Methylation process should be explained in details.
=>Since the main theme of this manuscript is the regulation of ELK3 expression level by miRNA, methylation process is beyond the scope of this manuscript.
Triple negativity of breast cancer should be elaborated in details
=>The key characteristic of TNBC is described at the first sentence of abstract (line 18-19).
In section of the Results if Authors use 5’-azacytidine for example they should explain in details role of the compound and present thet in the text or in the Fig 1 legend
=>We believe that the readers of Cancers are aware of detailed role of 5’-azacytidine. Nevertheless, we revised the manuscript with respect to reviewer’s comment.
(line 83-85) However, the expression level of ELK3 mRNA was not changed significantly following demethylation of the gene by the treatment of 5’-azacytidine, which inhibits DNA methyltransferase activity (Figure 1c).
Figure 3 and Figure 5 should be recombined in one figure and show together.
=>Figure 3 demonstrates that miR-200a directly targets 3’-UTR of ELK3 mRNA and Figure 5 demonstrates biological role of miR-200a/ELK3 axis. Figure 5 also provides mechanistic insights to explain the phenotype. For these reasons, we do not consider to combine figure 3 and figure 5.
miR-200a and ELK3, rest of unnecessary figures should be excluded
=>Since each figure has its own significance to support the results, we do not consider to exclude any of figures presented at figure 3 and figure 5.
Figure 5. Unnecessary figures should be excluded especially FGH parts because not clear , what Authors exactly want to show
=>Figure 5A-5C demonstrate that miR-200a/ELK3 axis regulates migration and invasion of MDA-MB231 in vitro. And figure 5D-5F provide mechanistic insights to explain the phenotype of figure 5A-5C. For these reasons, we do not consider to exclude any of figures presented at figure 5.
Figure 6. Unnecessary figures should be excluded
=>Figure 6A-6B demonstrate the biological role of miR-200a/ELK3 axis in vivo. Since each figure has its own significance to support the results, we do not consider to exclude any figures presented at figure 6.

Round 2
Reviewer 3 Report
Manuscript is not for publication
Authors reject to correct manuscript as recommended with explanation “we do not consider to exclude any of figures”
Authors are advised previously what to do with all figures, what needs more elaboration in the section of Introduction and Discussion
But For example how figure 5 should be reorganized
For example looking at figure 5
Figure 5A should be deleted because does not present anything significant and obvious related to cell migration or cell Invasion
Heat map should be explained in details, what gene is unregulated and what is down regulated
Fig 5C is only evidence if interaction of these two element but at gross level, it will be better if these experiment are shown in more details related to Methylation process, or phosphorilation process
Figure 5E Gene Ontology (GO) analysis is not derived from the experiment, but rather in general, which dos not have any meaning related to main funding
Figure 5F STRING analysis does not represent experimental confirmation of the manuscript but rather generated data hypothetical by computer program, it does not have any meaning in relation to the miR-200a and ELK3 or other genes experimentally found to be connected, so this figure should be excluded
It is the same with rest of Figures
Figure 4 A heat map correlation of miR-200a and ELK3 interaction for example Authors should show all significant gene up-regulated or down-regulated genes, however Fig 4B dos not have clear and any meaning related to main funding
Author Response
Dear Reviewer-3,
Thank you for your considerable comments about our manuscript.
Following the Academic Editor’s decision, we would like to leave main figures as they are.
With respect to comments of the reviewer about introduction and discussion, we revised our manuscript to include more elaborated explanation about several issues.
Detailed revision is described at attached file.
Sincerely yours,
Kyung-Soon Park

This manuscript is a resubmission of an earlier submission. The following is a list of the peer review reports and author responses from that submission.
Round 1
Reviewer 1 Report
Dear authors,
the content of the paper is good, good written, and the experiments are well designed and results well presented, although I would suggest to put the figures a bit bigger for better view and understanding.
I have some minor comments/suggestions:
Line 83: although the RT-PCR method is not a quantitative one, you could use a software to calculate a fold change on the mRNA levels, and be a bit more precise that saying "remarkably lower"...
Line 158: substitute "their" by the respective names.
Line 188: what is the difference between Figure 5a and Supplementary Figure 2?? DO you need the last? Does it add anything?
Figure 6A: title should be "48 hours after tail-vein injection" and not as it is on the paper
Line 236: you have a "-" too much.
Line 236 and 238: in silico, in vitro and in vivo should be in italic.
Line 248: "low expression of ZEB1 in MCF7 cells". Do you show it??
Line 249: cut off "presented here". It is redundant...
Lines 258-259: If ZEB1 is a target of miR-200a and transfection of the last did not lead to a decrease of ZEB1 levels, I would like to see those results. Why "data not shown"?? I think you should include those results.
Lines 265-267: not sure if you can state that...
Author Response
- Line 83: although the RT-PCR method is not a quantitative one, you could use a software to calculate a fold change on the mRNA levels, and be a bit more precise that saying "remarkably lower"...
We added the value of a fold change on the ELK3 mRNA levels in Fig 1D and revised the sentence.
(Line 86-87) The stability of the ELK3 mRNA was two-fold lower in MCF7 cells than in MDA-MB 231 cells at 480min after actinomycin D treatment.
- Line 158: substitute "their" by the respective names.
We changed the sentence to include respective names.
(Line 160-161) To extend our understanding of the negative association between miR-200a and ELK3 in breast cancer
- 3. Line 188: what is the difference between Figure 5a and Supplementary Figure 2?? DO you need the last? Does it add anything?
Figure 5a and figure 2S is the result of migration and invasion assay, respectively. To avoid confusion, we combined two figures and presented at Figure 5A.
- 4. Figure 6A: title should be "48 hours after tail-vein injection" and not as it is on the paper
We corrected all in Figure 6A and revised manuscript.
- Line 236: you have a "-" too much.
We checked “-” and removed “-”at the revised manuscript.
- Line 236 and 238: in silico, in vitro and in vivo should be initalic.
We checked in silico, in vitro and in vivo and changed in italic at the revised manuscript.
- Line 248: "low expression of ZEB1 in MCF7 cells". Do you show it ?
In the manuscript, we didn’t show the expression of ZEB1 in MCF7 cells. However, we published the data at our previous report (Cho HJ et al. Molecular Cancer Research, 2019).
- Line 249: cut off "presented here". It is redundant...
We removed the word in the sentence.
- Lines 258-259: If ZEB1 is a target of miR-200a and transfection of the last did not lead to a decrease of ZEB1 levels, I would like to see those results. Why "data not shown"?? I think you should include those results.
Thank you for your suggestion. We carefully repeated the experiment and we confirmed that ZEB1 mRNA level is decreased by the transfection of miR-200a, as reported in Wei J et al. 2014. We also confirmed that the decreased ZEB1 by miR-200a was not rescued by co-transfection of ELK3. We presented the data as supplementary figure 3 and revised the discussion of the manuscript.
(Line 294-300) Our data also showed that miR-200a suppressed ZEB1 mRNA level after miR-200a transfection, however introducing ELK3 was not rescued ZEB1 mRNA level in miR-200a transfected cells (Supplementary Figure 3), suggesting that the molecular link between miR-200a and ELK3 is not required for the function of ZEB1 as a transcriptional activator
- Lines 265-267: not sure if you can state that...
As described above, lines 265-267 was revised as lines 294-300 based on our carefully repeated experiment and reproducible data

Reviewer 2 Report
Summary: Accept after minor revision.
Kim et al. demonstrate that miR-200a negatively regulates the expression of ELK3 via the 3’-UTR of ELK3 mRNA. They then prove that the miR-200a/ELK3 axis controls cell invasion both in vitro and in vivo. Overall, the experimental design is thorough, methodical, and involves the proper controls. This study is well-executed, and this manuscript is well suited for Cancers.
Minor Critiques:
- How does miR-200a/ELK3 axis affect cell proliferation? Are the in vitro invasion results (Fig. 5A) fully decoupled from any potential effect on cell proliferation? The list of differentially expressed genes (Fig. 5C) does contain genes like CCNE1 which may affect proliferation.
- A brief description of what genes regulate miR-200a levels would be beneficial in the discussion section.
- Fig. 3B contains no bar for MDA. Please add this.
Author Response
- How does miR-200a/ELK3 axis affect cell proliferation? Are the in vitro invasion results (Fig. 5A) fully decoupled from any potential effect on cell proliferation? The list of differentially expressed genes (Fig. 5C) does contain genes like CCNE1 which may affect proliferation.
For the invasion assay, we transfected cancer cells with miR-200a/ELK3 for 48 h, and then 5 x 104 of transfected cells were transferred to the transwell inserts (Day 0). After 24 h (Day 1), invasive cells were analyzed. To examine whether in vitro invasion results (Fig. 5A) was affected by cell proliferation, we performed cell counting assay in the time table of the invasion experiment. Cell numbers were not significantly different between control (NC), miR200a and miR200a/ELK3 group at Day 1. Based on these results, we suggest that invasion results are decoupled from potential effect on cell proliferation in spite of that CCNE1 is contained in the result of RNA Seq analysis.
- A brief description of what genes regulate miR-200a levels would be beneficial in the discussion section.
Following the reviewer’s suggestion, we revised the manuscript to describe the regulator of miR-200a.
(Line 294-296) It is well-known that ZEB1 and miR-200 forms feedback loop, and the miR-200a/ZEB1 axis has been recognized as a determinant of cellular plasticity during cancer progression [27-30].
We also include the potential target of miR-200a/ELK3 axis in the discussion.
(Line 300-305) Interestingly, we found that miR-200a/ELK3 axis regulates the expression of Lama5 (Lamininα5) and TNS1 (Tensin1), which encode extracellular proteins [31]. This founding provides the possibility that miR-200a negatively regulates ELK3-mediated cell migration and invasion through extracellular matrix remodeling. Our finding is consistent with previous reports that one of major function of microRNAs (miRNAs) is to regulate extracellular matrix-related gene to control cancer invasiveness [32, 33].
- Fig. 3B contains no bar for MDA. Please add this.
Basically, the miR-200a expression is huge different between MDA-MB 231 and MCF-7 cells. We changed the graph pattern to clearly show the difference.

Reviewer 3 Report
The manuscript by Kim et al. describes the role of the miR2001/ELK3 axis in the regulation of metastatic phenotype of TNBC subtypes. Overall, the experimenst have been weel performed and the study adds new mechanistic information on how ELK3 partecipates to breast cancer aggressive phenotype. Here, I detailed few concerns that if addressed could improve the quality of the manuscript.
- The authors analyzed the methylation status of CpG islands within the first intron. Why did they studied intron 1 and not the canonical promoter region? UCSC browser highligted the existance of CpG islands both at the canonical promoter upstream of exon 1 and within intron 1. The rationale of this analysis should be better explained. Which is the statistical significance of methylation study results?
- 3A and Suppl. Fig. 1B lack statistical analysis.
- Which information Fig. 3A adds to the already described Fig.3C? If I’m not wrong, Fig.3C represents the same experiment shown in Fig.3A with miR-200A uniquely.
- In addition to ERK3 mRNA levels, protein expression levels should be analyzed in the same cells transfected with NC mimic, miR200a and miR200a+ELK3.
- RNA-seq analysis should be deposited. In addition, gene categories analysis has to be performed (as GO, KEGGS pathways…). Which are the main biological pathways affected by ELK3/miR200a axis deregulation? Validation of DEGs has to be done by RT-qPCR and for EMT gene salso by protein quantification. ChIP analysis of ELK3 binding to at least to Cdh1 target gene before and after miR200a transfection.
- Figure 6A: quantization of whole-body bioluminescence in control and treatment groups has to be shown. The images can not easily understood: lung imaging should be improved and total animal imaging should allow to better appreciate the localization of tumor cells in the lung exclusively. The expression levels of ELK3 in GFP cells found in the lungs have to be analyzed.
- The quality of the graphical abstract should be improved with the addition of some biological details, such as gene targets.
Author Response
Reviewer #3.
- The authors analyzed the methylation status of CpG islands within the first intron. Why did they studied intron 1 and not the canonical promoter region? UCSC browser highligted the existance of CpG islands both at the canonical promoter upstream of exon 1 and within intron 1. The rationale of this analysis should be better explained. Which is the statistical significance of methylation study results?
We agree with the reviewer’s comment. Indeed, we tried very hard to analyze the methylation status of CpG islands within the canonical promoter region. Unfortunately, we could not amplify the promoter regions (Promoter-1 and Promoter-2) even though we performed PCR with two different primer sets under various conditions. We think that the failure of PCR might be caused by GC-rich in the region. As an alternative, we analyzed intron-1, which are reported to be linked with gene expression (Anastasiadi, D et al. 2018).
* Anastasiadi D et al. Consistent inverse correlation between DNA methylation of the first intron and gene expression across tissues and species. Epigenetics Chromatin. 11(1):37 (2018)
- 3A and Suppl. Fig. 1B lack statistical analysis.
We analyzed statistical analysis and added p-value in the legend and figure of supplementary figure 1B.
- Which information Fig. 3A adds to the already described Fig.3C? If I’m not wrong, Fig.3C represents the same experiment shown in Fig.3A with miR-200A uniquely.
Thank you for your comment. Since figure 3a and figure 3c represents the same result, we deleted figure 3c and the sentence in the revised manuscript.
- In addition to ERK3 mRNA levels, protein expression levels should be analyzed in the same cells transfected with NC mimic, miR200a and miR200a+ELK3.
We added ELK3 protein level as figure5C.
- RNA-seq analysis should be deposited. In addition, gene categories analysis has to be performed (as GO, KEGGS pathways…). Which are the main biological pathways affected by ELK3/miR200a axis deregulation? Validation of DEGs has to be done by RT-qPCR and for EMT gene salso by protein quantification. ChIP analysis of ELK3 binding to at least to Cdh1 target gene before and after miR200a transfection.
Following the reviewer’s suggestion, we deposited our RNA-seq data (GSE148562). We also analyzed the role of miR200a/ELK3 on biological pathways using GO analysis. We confirmed the target genes of miR200a/ELK3 using qRT-PCR. All new data were presented as figure 5d-5g and manuscript was revised to include the result.
(Line 193-206) To further examine the molecular mechanisms involved in the regulation of cell migration and invasion by the miR-200a/ELK3 axis, we analyzed the gene expression profiles of miR-200a alone or miR-200a plus pcDNA3.1-Flag-ELK3 compared with negative control in MDA-MB 231 cells. An analysis of differentially expressed genes (DEGs) showed that the negative control group and miR-200a plus ELK3 group had similar genetic signatures compared with the miR-200a alone group (Figure 5d). Gene Ontology (GO) analysis reveals that 99 DEGs were significantly enriched in membrane and extracellular-related components, and cell migration- related pathways (Figure 5e). Notably, STRING analysis reveals that CDH1 is a hub of 99 DEGs (Figure 5f). Combining with a previous study that ELK3 directly represses CDH1 expression in MDA-MB 231 [17], this result indicates that CDH1 might be the main target of miR-200a/ELK3 axis to regulate migration and invasion of cancer cells. To validate the speculation, we performed qRT-PCR analysis of 8 genes including CDH1. Along with CDH1, Lama5 (Lamininα5) and TNS1 (Tensin1), which encode extracellular proteins, was confirmed to be regulated by miR-200a/ELK3 axis (Fig. 5G).
- Figure 6A: quantization of whole-body bioluminescence in control and treatment groups has to be shown. The images can not easily understood: lung imaging should be improved and total animal imaging should allow to better appreciate the localization of tumor cells in the lung exclusively. The expression levels of ELK3 in GFP cells found in the lungs have to be analyzed.
Thank you for your comment. The purpose of Figure 6a experiment was to analyze the role of miR200a/ELK3 axis on the extravasation ability of cancer cells. Therefore, we sacrificed mice 48 h after tail-vein injection of control and indicated samples. Unfortunately, the bio-luminescence signal in whole body was not strong enough to quantify at 48 h after tail-vein injection.
Following the reviewer’s suggestion to improve lung image, we changed Figure 6B to show more clear GFP-positive cells in the lung tissue
To further detect ELK3 in GFP cells, we performed immunofluorescence staining and immunoblot analysis of ELK3 protein, and qRT-PCR analysis of ELK3 mRNA with the lung tissues. However, either of three methods was not sensitive enough to detect ELK3 signal in the lung tissue. We believe that the number of extravasated GFP (+) cells occupies a small amount in the lung tissue at our experimental time-frame (48 h after tail-vein injection).
- The quality of the graphical abstract should be improved with the addition of some biological details, such as gene targets.
We improved the graphical abstract to emphasize the role of miR200a/ELK3 axis during metastasis.

Reviewer 4 Report
Manuscript is not for publication
Abstract should be more clear and concise
Figure 1 is not clear. Authors should explain what purpose of the experiment is
Figure 2 It is not clear what Authors wants to show in this figure, the effect of actinomycin only?
Authors need to show experiments of ELK3 and mir-200a correlation like 3A Figure
All figures should be rearranged with intention to show link between ELK3 and mir-200a
Figure 3 needs detailed explanation of 3’UTR of ELK3 luciferase in Humans
Figure 4 and 6 . Heat map from these two figures should be rearranged and gene involved in details explained. And cell invasion figure should be combined with Figure 6
Unnecessary figure from Fig 6b should be excluded (first par of Fig 6b)
In introduction and in Discussion Authors need to elaborate more on the link between ELK3, mRNA200a with major cancer signaling molecule AKT involved in metastasis
Author Response
Reviewer 4.
- Manuscript is not for publication
In this manuscript, we report a novel post-transcriptional regulatory mechanism that determines ELK3 expression level in breast cancer subtypes. We found that the expression level of miR-200a and ELK3 mRNA are negatively correlated in the luminal and triple-negative breast cancer cells and miR-200a targets 3’UTR of ELK3. We also demonstrated that miR-200a and ELK3 expression is functionally linked to regulate the metastatic characteristics of breast cancer cells. We believe that our work provides the novel insight to understand the mechanisms underlying the link of ELK3 and miR200a regulates breast cancer metastasis. Such as above reasons, we believe that this study will be of considerable interest to the readers of Cancers.
- Abstract should be more clear and concise
We improved the quality of the graphical abstract to provide biological details of functional link between miR-200a and ELK3 that regulates metastatic nature of breast cancer.
- Figure 1isnot clear. Authors should explain what purpose of the experiment is
We changed the sentence in the manuscript (line 79-81) to show clearly the purpose of the experiments. All data in Figure 1 showed the difference of ELK3 mRNA level in luminal type and basal type breast cancer cell line is determined by mRNA stability, not by methylation of ELK3 regulatory gene.
- 4. Figure 2 It is not clear what Authors wants to show in this figure, the effect of actinomycin only?
To elucidate the underlying mechanism to determine mRNA stability of ELK3, we used actinomcyin D which is an mRNA transcription inhibitor. After actinomycin D treatment, endo-ELK3 showed rapid degradation of ELK3 mRNA compared with exo-ELK3 without 3’UTR, suggesting 3’UTR determines the stability of ELK3 mRNA.
- Authors need to show experiments of ELK3 and mir-200a correlation like 3A Figure
We showed the correlation of ELK3 and miR-200a in Figure 3A and 3C as well as Figure 4. These figures that demonstrated the ELK3 and miR-200a showed negative regulation in breast cancer and breast cancer patients.
- All figures should be rearranged with intention to show link between ELK3 and mir-200a
We demonstrated that the link between ELK3 and miR-200a in the breast cancer metastasis as following order: 1) in vitro experiment (Figure 1-3) showed miR200a negative regulates ELK3 mRNA level through the binding of 3’UTR. 2) Breast cancer cell lines and breast cancer patient sample analysis (Figure 4) showed the negative correlation between miR200a and ELK3. The results confirmed our in vitro data. 3) miR200a/ELK3 axis has a biological function to regulate migration and invasion of cancer cells in vitro and in vivo (Figure 5 and Figure 6). We are convinced that our figures are well-organized to support the conclusion of this manuscript.
- Figure 3 needs detailed explanation of 3’UTR of ELK3 luciferase in Humans
We used human breast cancer cell line MDA-MB 231 to analysis the transcriptional activity of ELK3 by luciferase analysis. As shown Figure 3D, 3’UTR of ELK3 conserved among all species including human.
- Figure 4 and 6. Heat map from these two figures should be rearrangedand gene involvedin details explained. And cell invasion figure should be combined with Figure 6. Unnecessary figure from Fig 6b should be excluded (first par of Fig 6b)
We have two heat map data in Figure4 and Figure 5 (old version of figures). The one in Figure 4a showed that the expression pattern in miR-200a and ELK3 in all breast cancer cells, suggesting particularly luminal and normal-like breast cancer cells showed negative correlation between miR-200a and ELK3 expression. The other one in Figure 5d (revised version of figures) showed that the differential expression genes (DEGs) in MDA-MB 231 cells with negative control, miR-200a, and miR200a plus ELK3. So they are different set of gene analysis and purpose.
We also analyzed the role of ELK3/miR200a on biological pathways using GO analysis. We confirmed the target genes of miR200a/ELK3 using qRT-PCR. All new data were presented as figure 5d-5g and manuscript was revised to include the result.
(Line 193-206) To further examine the molecular mechanisms involved in the regulation of cell migration and invasion by the miR-200a/ELK3 axis, we analyzed the gene expression profiles of miR-200a alone or miR-200a plus pcDNA3.1-Flag-ELK3 compared with negative control in MDA-MB 231 cells. An analysis of differentially expressed genes (DEGs) showed that the negative control group and miR-200a plus ELK3 group had similar genetic signatures compared with the miR-200a alone group (Figure 5d). Gene Ontology (GO) analysis reveals that 99 DEGs were significantly enriched in membrane and extracellular-related components, and cell migration- related pathways (Figure 5e). Notably, STRING analysis reveals that CDH1 is a hub of 99 DEGs (Figure 5f). Combining with a previous study that ELK3 directly represses CDH1 expression in MDA-MB 231 [17], this result indicates that CDH1 might be the main target of miR-200a/ELK3 axis to regulate migration and invasion of cancer cells. To validate the speculation, we performed qRT-PCR analysis of 8 genes including CDH1. Along with CDH1, Lama5 (Lamininα5) and TNS1 (Tensin1), which encode extracellular proteins, was confirmed to be regulated by miR-200a/ELK3 axis (Fig. 5G).
In addition, we combined the cell invasion and migration data, and presented it as Figure 5A.
Figure 6b is an essential image to support our conclusion by showing extravasated tumor cells that are localized in the lung tissue.
- In introduction and in Discussion Authors need to elaborate more on the link between ELK3, mRNA200a with major cancer signaling molecule AKT involved in metastasis
In this manuscript, we demonstrated that miR-200a directly target to 3’-UTR of ELK3 mRNA and miR-200a/ELK3 axis is functionally linked to regulate breast cancer metastasis. Our view on this issue is that signaling pathway involved in metastasis is beyond the scope of this manuscript. Furthermore, our unpublished data showed know-down of ELK3 does not disturb AKT signaling of MDA-MB231 cells.
